# Evaluating the Suitability of 3D Bioprinted Samples for Experimental Radiotherapy: A Pilot Study

**DOI:** 10.3390/ijms23179951

**Published:** 2022-09-01

**Authors:** Munir A. Al-Zeer, Franziska Prehn, Stefan Fiedler, Ulrich Lienert, Michael Krisch, Johanna Berg, Jens Kurreck, Guido Hildebrandt, Elisabeth Schültke

**Affiliations:** 1Department of Applied Biochemistry, Institute of Biotechnology, Technische Universität Berlin, 13355 Berlin, Germany; 2Department of Radiooncology, Rostock University Medical Center, 18059 Rostock, Germany; 3European Molecular Biology Laboratory (EMBL), Hamburg Outstation/DESY, 22607 Hamburg, Germany; 4PETRA III/DESY, Beamline P21.2, 22607 Hamburg, Germany; 5European Synchrotron Radiation Facility (ESRF), 38043 Grenoble, France

**Keywords:** 3D bioprinting, DNA damage, gammaH2AX, experimental radiotherapy, high dose rate radiotherapy, human lung cancer cells, microbeam radiotherapy (MRT)

## Abstract

Radiotherapy is an important component in the treatment of lung cancer, one of the most common cancers worldwide, frequently resulting in death within only a few years of diagnosis. In order to evaluate new therapeutic approaches and compare their efficiency with regard to tumour control at a pre-clinical stage, it is important to develop standardized samples which can serve as inter-institutional outcome controls, independent of differences in local technical parameters or specific techniques. Recent developments in 3D bioprinting techniques could provide a sophisticated solution to this challenge. We have conducted a pilot project to evaluate the suitability of standardized samples generated from 3D printed human lung cancer cells in radiotherapy studies. The samples were irradiated at high dose rates using both broad beam and microbeam techniques. We found the 3D printed constructs to be sufficiently mechanically stable for use in microbeam studies with peak doses up to 400 Gy to test for cytotoxicity, DNA damage, and cancer cell death in vitro. The results of this study show how 3D structures generated from human lung cancer cells in an additive printing process can be used to study the effects of radiotherapy in a standardized manner.

## 1. Introduction

With an incidence between 33.9 and 60.0 cases per 100,000 people per annum, malignant lung tumours belong to group of the most common cancers worldwide (World Cancer Research Fund). Of all cancer entities, lung cancer has the highest incidence and death rate globally, with a prognosticated increase within the next twenty years [1]. Therefore, the development of new therapeutic concepts which improve the quality of life and overall survival for patients with lung cancer are of high priority.

Radiotherapy is an important therapy component for many lung cancer patients. Given the limited clinical success of current therapeutic concepts, new radiotherapy techniques promising improved tumour control with the same or even better normal tissue tolerance are currently undergoing preclinical evaluation. These new irradiation concepts include high dose rate irradiation techniques, such as FLASH radiotherapy and microbeam radiotherapy (MRT). Both concepts are characterized by an ultrafast dose deposition in the irradiation target, taking advantage of the FLASH effect, leading to an excellent normal (non-tumour) tissue preservation when X-ray energy is deposited at dose rates in excess of 40 Gy/s [2].

While FLASH radiotherapy is a broad beam technique, the MRT concept is based on spatial dose fractionation at the micrometre range, in addition to a high dose rate. A multislit collimator is inserted in the path of the incident beam, resulting in an inhomogeneous dose distribution in the irradiation target. Dose deposition in the irradiation target is characterized by a periodic sequence of high dose (peak dose) zones and low dose (valley dose) zones. Because the dose is deposited only in a very small volume, the prescribed peak doses can be higher by one or even two orders of magnitude, compared to current clinical radiotherapy approaches. Those high peak doses combine with bystander effects generated in neighbouring tissue to control tumours very effectively, as has been shown in small animal models of malignant brain tumours [3].

Until recently, MRT has almost exclusively been tested in small animals, in order to understand the effects of microbeams on normal brain tissue and in brain cancer [4,5,6]. More recently, however, there has been a growing interest in whether MRT could also benefit patients with lung cancer [7,8,9]. At this stage of MRT development, no statement can yet be made regarding the tumour stage (TNM status) at which it could be successfully integrated into a clinical radiotherapy schedule.

In order to compare outcomes between different treatment modalities and in different technical setup scenarios, it is necessary to devise a set of standardized samples for use as controls. Over the last decade, 3D bioprinting techniques have been developed which allow the relatively simple production of 3D structures [10,11] and tumour models [12]. Bioprinted models have been used in biomedical research efforts, including disease modelling, regenerative medicine, tissue engineering, infectious diseases studies, and cancer research [12,13]. Compared to the still widely used 2D cell cultures, this technology offers significant advantages, and it has been used in medical imaging studies [14,15,16] and as platform to test antineoplastic drugs in vitro [17,18]. The 3D models are superior to the 2D cell cultures because cells are positioned within a substrate at a precise location reflecting the particular design of the experiment. With 3D intercellular interactions, these structures are much more similar to spontaneous human tumours than 2D cell cultures [19,20,21].

In experimental radiotherapy at the synchrotron, in vitro samples are exposed to stresses which are untypical for the cell culture environment, including heat development and tension, when irradiated in other than horizontal positions. This could potentially impact both the structural integrity and the viability of the samples. In order to develop standardized 3D samples suitable for work in experimental radiotherapy, we have designed a study with tumour samples generated by a 3D bioprinting process using human lung cancer cells that mimic a tumour in a microenvironment. Studies on the characteristics of hydrogel matrices produced using the 3D bioprinting technique have been conducted previously in RSC96 cells (a rat Schwann cell line) and human umbilical vein endothelial cells (HUVECs) [22,23].

For our study, a different hydrogel basis and the A549 cell lung cancer cell line were chosen because this combination had been previously shown to be well suited for bioprinting approaches based on a bio-ink composed of gelatin, alginate, and Matrigel [11], and lung cancer is of interest in MRT studies. The bioprinted 3D structures were exposed to different doses of high dose rate radiotherapy using both broad beam and microbeam techniques and then analysed for metabolic activity, cell death, and DNA damage. We demonstrate that the high dose rate radiotherapy induces cell death specifically, and cell death is limited to the site of irradiation, similar to the results described in tissue samples from animal studies on lung carcinoma [8]. This is especially impressive after microbeam irradiation with peak doses of several hundred Gy. While a high number of DNA double-strand breaks is detected within the paths of the microbeams, in the valley dose zones between the microbeams, within a distance of only a few micrometres, few to no DNA double-strand breaks are seen. This phenomenon is accompanied by a decrease in metabolic activity and an induction of DNA damage in the irradiated samples, which is very different from what is seen in the naïve controls. This type of analysis would be extremely difficult, if not impossible, to conduct in 2D cell cultures, since in the latter, cells typically migrate and mix between irradiated and non-irradiated zones within hours after irradiation. Subsequently, cells are no longer identifiable as directly irradiated (in the path of the microbeams) vs. not directly irradiated (in the valley dose zones during the irradiation process). This study can be considered as a first step towards the optimization of 3D models in experimental radiotherapy.

## 2. Results

### 2.1. Setup and Mechanical Stability of the 3D Printed Tumour Samples

The A549 cell line, a KRAS mutant and an EGFR wild type epithelial carcinoma obtained from a 58-year-old male patient, is one of the most commonly used cell lines to model non-small cell lung cancer (NSCLC). Roughly 85% of all lung cancer cases are of this type, making it the world’s most common cause of cancer-related fatalities. To model lung cancer for radiotherapy experiments, A549 cells were printed into a simple 3D grid-like structure using a gelatin-alginate-Matrigel bio-ink, as previously described [11].

Irradiation was carried out at beamline P21.2 of the PETRA III synchrotron (DESY, Hamburg, Germany). The 3D bioprinted constructs, with dimensions of approximately 10 × 10 × 2 mm^3^, were placed into microwell plates, covered with only a thin film of growth medium, and mounted in a vertical position during the irradiation procedure. The mechanical support for the irradiation of the 3D printed matrices, with and without cells, is shown in Figure 1a. During the exposures, the samples were translated through the microbeam array at a constant speed, creating a regular vertical stripe pattern, as shown on the radiochromic film attached to the back of the plate (Figure 1b). The straightness and homogeneity of the stripes on the films provide evidence of the constant and vibration-free motion of the vertical stage (Figure 1c). The samples were mechanically stable and remained in position during exposure.

For irradiation, the samples were moved vertically through the beam at a speed of 1.2 mms/s for an MBI peak dose of 40 Gy, and at a speed of 0.12 mm/s for an MBI peak dose of 400 Gy, respectively. The doses were chosen to reflect an earlier in vivo experiment in which the normal tissue tolerance of lung tissue was studied [7]. In this study, a field of 3.9 × 13 mm in the right lung of healthy adult CL57/BL6 mice was irradiated with MRT peak entrance doses of 40 Gy and 400 Gy; the valley doses were 0.42 Gy and 4.2 Gy, respectively. No signs of dyspnea or distress were observed in the animals within the first three days after irradiation, although traces of the microbeams in the harvested lung tissue were clearly visible as DNA double-strand breaks when immunostained with gamma H2AX.

### 2.2. Dose Recording

Based on our measurements, a dose rate of about 32 Gy/s in the microbeam peaks was determined and approximately 40 Gy and 400 Gy was administered, respectively. The intensity profiles recorded on the Gafchromic™ films (Figure 2), when compared to the non-linear calibration curves, verified that the administered doses were in the expected dose range. It should be noted that, due to the extremely steep dose fall-off at the edges of the microbeams, the only significant difference between the beam profiles recorded after irradiation of the films with 40 Gy and with 400 Gy is the beam intensity, correlating to the extent of dose entry.

### 2.3. Analysis of Cell Viability and Metabolic Activity after Radiotherapy

Cell viability was qualitatively assessed 36 h post-irradiation by examining living (calcein-AM, green) and dead (ethidium homodimer-1, red) cells under a microscope. Cell viability was high in all samples, as shown in Figure 3a. In comparison to the control (no radiotherapy, the bioprinted models were placed upright for an equivalent amount of time in an environment of comparable heat and humidity), the samples exposed to microbeam peak doses of 400 Gy had a higher number of dead cells in comparison with the 40 Gy and control unexposed samples. The lines of red fluorescing dead cells after irradiation with 400 Gy reflects the beam geometry recorded on the HD-V2 Gafchromic™ films mounted under the 24-well plates (Figure 3a). Then, the percentages of living and dead cells were calculated using ImageJ (Figure 3b). Interestingly, around 5% of printed A549 cells were dead in the control sample, and only the highest irradiation dose, 400 Gy, led to an obvious increase in dead cells to 21% (Figure 3b). In contrast, the 40 Gy irradiation had lower toxic activity and resulted in death of almost 10% of the printed cells (Figure 3b).

On days two, five, and ten following the irradiation, the metabolic activity of the printed A549 cells was assessed by measuring the reduction in the tetrazolium salt XTT to formazan by dehydrogenase enzymes. The metabolic activity remained high throughout the entire time course of the experiments in the control samples, but declined significantly on day two after 40 Gy and 400 Gy of irradiation (Figure 3c). This reduction in viability was also observed on days five and ten after irradiation, and was more pronounced in samples irradiated with 400 Gy compared to 40 Gy (Figure 3c).

### 2.4. Radiotherapy Induces DNA Double-Strand Breaks in A549 Printed Cells

We investigated γH2AX as a marker of the DNA damage (DNA double-strand breaks) in A549 printed structures after irradiation with two different microbeam peak doses (40 Gy and 400 Gy). We demonstrated that γH2AX levels increased in a dose-dependent fashion after irradiation, as determined by immunofluorescence and immunoblot analysis, respectively. Figure 4a demonstrates that the γH2AX fluorescence intensity increased in printed cells irradiated with 40 and 400 Gy. To further investigate the correlation between the dose intensity and DNA damage induction, we subjected the A549 cells irradiated with 40 and 400 Gy to immunoblot analysis. The γH2AX protein level was increased significantly in samples exposed to 400 Gy compared to controls, and also to a lesser extent in samples irradiated with microbeam peak doses of 40 Gy, indicating that the dose is correlated with the intensity of the γH2AX, as detected by immunoblots (Figure 4b,c).

## 3. Discussion

In clinical radiotherapy, parameters such as progression-free survival (PFS) and overall survival (OS) are used in the qualitative evaluation of therapeutic success. In the preclinical development phase, however, very few standards exist. PFS and OS are frequently studied in small animal models of human disease. Due to significant differences in the normal life expectancy between small animals and humans, however, it is difficult to predict absolute time gained in human patients based on the results obtained in small animal models. Concordance between animal models and clinical trials is still difficult to achieve, with an average percentage of concordant outcomes of less than 8% [24]. Furthermore, ethical concerns about animal studies have created a trend towards replacement methods, where in vivo studies are supplemented and partially replaced by suitable in vitro experiments. The need to build cellular models that better depict the complexities of tumour biology is driving the shift from 2D cultures to 3D cultures. Traditional cancer models, such as 2D cell culture methods and 3D cancer spheroids, appear to be deficient in essential tumour microenvironment components. The use of 3D bioprinting, on the other hand, offers user-controlled deposition of supporting matrix biomaterials, cells, and biomolecules in a predefined architecture, as well as an opportunity for the cells to connect and generate 3D constructs which are more similar to in vivo tissues than 2D cell cultures [25]. While the environment in 3D bioprinted structures is still less complex than in in vivo organisms, the environment can be modified to suit the requirements of preliminary studies in a standardized situation, thus helping to reduce the number of experimental animals. It seems reasonable to develop biological standard samples which can be utilized, similar to a common denominator, with any (or at least many) different therapy approaches to compare the outcomes.

While 3D bioprinted constructs are already in use in pharmacology [26] and infectious disease studies [11], this was the first evaluation of 3D bioprinted constructs in high dose rate radiotherapy. The printing strategy used in this study is suitable because it allows for optimization of the bio-ink so that the printed structures are mechanically stable, despite the temporary heat development during the irradiation at high dose rates.

In the past, 2D cell cultures have been indispensable in understanding cell and molecular biology, disease mechanisms, and tissue engineering. They are still frequently utilized in preclinical drug research, cancer research, and genomics studies. Although 2D monolayer cell cultures provide valuable information on basic tumour biology and radiobiology, they do not accurately reflect the complexity of 3D solid tumours. The value of 2D cultures in microbeam experiments has been shown to be limited because many cells migrate out of the path of the microbeams within less than 24 h after irradiation, and the geometrical pattern typical for microbeam irradiation is subsequently lost [27].

During the last decade, significant progress has been made in optimizing the mechanical characteristics of bio-inks for lung tissue structures. Bio-inks based on an extracellular matrix produced from decellularized lung tissue have been developed to mimic the biochemical composition of normal lung tissue [28]. The focus has been on the tracheal region, including epithelization and cartilage development after implantation in animal models [29]. Another major area of research is the construction of 3D lung models to examine the blood–air barrier and alveoli, which are important in modelling lung tissue function [30,31]. With cell migration and invasion comparable to that seen in real tissue for at least four weeks, the structural and analytic value of 3D models surpasses that of 2D cultures [32]. None of these studies, however, used bioprinted structures to assess the effects of irradiation with therapeutic intent. We have chosen the A549 lung cancer cell line for our high dose rate radiotherapy studies at the PETRA III Synchrotron in Hamburg because its printability had been previously well documented. We evaluated the sensitivity of the printed A549 cells toward radiotherapy. Using the live/dead stain at 36 h after microbeam irradiation, we were able to show that the majority of irradiated cells were still present in the path of the microbeams. However, these cells were not viable anymore, unlike the cells in the sham-irradiated cells within the same type of 3D constructs. This points towards an early (acute) and locally well-defined effect of cancer cell destruction in the paths of the microbeams. While this observation has previously been made in vivo, this is the first time that the same effect was shown in vitro. Contrary to 2D cell cultures, the paths of the microbeams were still clearly defined at 36 h after microbeam irradiation. Similar to tissue from in vivo studies, the number γH2AX foci was significantly higher in irradiated samples, compared to non-irradiated samples. We have developed the technical prerequisites for an in vitro model as the basis for further structural and molecular biology studies in experimental radiotherapy in order to develop a better understanding of the cellular response in both the tumour and the normal tissue environment. To achieve the latter, it is necessary to develop 3D models which include the tumour enclosed in a normal tissue matrix.

In this study, 3D bioprints have been tested exclusively with its future use in basic experimental research in mind, using a commercially available human lung cancer cell line to produce standard samples. However, as an approach to individualized medicine, it would be possible to use cell lines derived directly from human primary lung cancer samples to generate 3D bioprints and test a number of potentially suitable cytotoxic agents, including compounds for chemotherapy and immunotherapy, in combination with radiotherapy, to identify the most suitable therapy scheme.

## 4. Materials and Methods

### 4.1. Cell Culture and 3D Bioprinted Constructs Preparation

In this study, we used a microextrusion bioprinter (Incredible and BioX (Cellink, Göteborg, Sweden) to generate 3D constructs in which spaced pores were arranged in a grid-like pattern (length 1 cm × width 1 cm × height 0.1 cm). This pattern is widely used in 3D bioprinting applications because the pores allow printed cells to be supplied with nutrients and oxygen during cultivation, unlike completely closed 3D constructs. Human epithelial lung cancer cells (A549, ATCC, Manassas, VA, USA) were grown in DMEM high glucose (Biowest, Nuaillé, France) with 10% fetal bovine serum (FBS; c.c.pro, Oberdorla, Germany), 1% 100X L-Glutamine (Biowest), and 1% penicillin/streptomycin (P/S; Biowest). For the preparation of the bio-ink, gelatin powder (Sigma, Steinheim, Germany) was dissolved in DMEM high glucose at 37 °C on a magnetic stirrer. After 2 h, sodium alginate powder (Sigma) was added and stirred constantly overnight at 37 °C. The final cell-laden bio-ink contained alginate (2% *w*/*v*, Sigma), gelatin (3% *w*/*v*, Sigma), Matrigel (20% *v*/*v*, Corning, NY, USA), CaSO_4_ (0.03 M, Roth, Karlsruhe, Germany), and A549 cells (7 × 10^6^ cells/mL), as described previously [10]. The suspension was printed in three layers. We plan to increase the thickness of the 3D prints in the future to increase the similarity with in vivo tissues. After printing, the samples were submerged in 100 mM CaCl_2_ to increase the polymerization of the alginate. Subsequently, they were covered with growth medium (DMEM high glucose, supplemented with 10% FCS, 2 mM glutamine and antibiotics) and stored in a standard incubator at 37 °C and 5% CO_2_.

### 4.2. Radiation Source and Measurement of Irradiation

All irradiations were carried out in the experimental hutch/experimental station at the synchrotron with beamline P21.2 at PETRA III. The radiation source of P21.2 is an in-vacuum undulator located a 150 m distance from the experiment. A bent-Laue monochromator produces a monochromatic high energy beam with a relative bandwidth of 10^−3^ at 45 keV, corresponding to the 6th harmonic of the undulator’s fundamental energy. The low divergence of the undulator source at a large distance allows for the acquisition of a quasi top-hat beam profile of 3.6 × 1.5 mm^2^ (horizontal × vertical). Flux measurements have been carried out with a partially depleted silicon diode (Canberra PIPS PD500) calibrated by the German Federal Office of standards (PTB Berlin). Under these conditions, 6.29 × 10^12^ ph/s were delivered.

Irradiation studies were conducted using both spatially unfractionated (broad) beam and microbeam irradiation (MBI) techniques. For MBI, a fixed-space multi-slit collimator (UNT, Morbier, France) was used to split the incident beam into an array of quasi-parallel microbeams. The width of each individual microbeam was 50 µm and the centre-to-centre distance between the microbeams was 400 µm. The irradiation target was hit with an array of a total of 9 microbeams.

The samples were transported between the laboratory and the beamline in 24-well plates in a mobile incubator (Cellbox Flight™, Hamburg, Germany). For irradiation, most of the growth medium was aspirated so that only a thin film of fluid was left in each well. In order to limit the variation of energy deposition due to sample depth, the sample was positioned to face the beam in its smallest diameter. The 24-well plates were fixed upright into a mechanical support facing the beam, with the samples sticking to the bottom of the wells (Figure 1). Immediately after irradiation, fresh medium was added to the samples. The duration of the irradiation process itself is on the order of seconds. However, including mounting, making the experimental hutch safe for irradiation, and recovering the sample after irradiation, the entire procedure took approximately five minutes.

Contrary to clinical radiotherapy, where the gantry is rotated around the patient, the position of the synchrotron beam is fixed and cannot be moved across the sample. If the height of the irradiation target is larger than the vertical beam height, the object or subject to be irradiated needs to be translated through the beam vertically. The samples were mounted on a vertical translation stage with adjustable speed downstream of the multislit collimator.

Self-developing EBT 3 and HD-V2 Gafchromic™ film (Ashland, Wilmington, DE, USA), mounted behind the samples, was used to record and verify that the irradiation pattern corresponded to the intended irradiation geometry, and that the target dose was in the intended dose range. The spatial resolution is approximately 25 µm for EBT3 and approx. 5 µm for HD-V2 film, which makes them well-suited for the registration of microbeams. The dynamic dose ranges are 0.1–20 Gy for EBT3 film and10–1000 Gy for HD-V2 film, allowing for dose determination for the peak doses (HD-V2) and for the valley doses (EBT3). The energy dependence of the film, mostly quoted as less than 3%, is minimal. High resolution scans (10,000 dpi) were obtained from the exposed films and analysed using the Histogram function of ImageJ [33]. Mean intensity values were obtained and correlated to two types of calibration curves. One calibration curve was obtained at the synchrotron source, based on measurements actually obtained locally, and one calibration curve was obtained from films exposed at a conventional orthovoltage source at 100 kV, which is periodically calibrated for the treatment of human patients according to the PTB standards (X-beam, Varian). The latter was created for the purpose of fine tuning, because the calibration curve obtained at the synchrotron contained only 6 dose values and therefore, allowed only a coarse assessment of the dose obtained within a single microbeam.

### 4.3. Processing and Immunostaining after Irradiation

Bioprinted constructs were fixed in 3.7% formalin (Sigma), embedded in paraffin, and sliced into 10 μm thick sections. After deparaffinization, antigen retrieval was performed in Tris-EDTA (10 mM Tris Base, 1 mM EDTA, pH 9.0) at 95 °C for 30 min. Cells were permeabilized using 1% Triton-X-100 for 15 min, and blocking was performed for 30 min using 5% goat serum. For the characterization of nuclear damage, γH2AX (anti-γH2A.X (phosphor S139), Abcam, ab11174, 1:1000, Berlin, Germany) was used. For the cellular characterization of epithelial cells, pan-cytokeratin (anti-pan Cytokeratin, abcam, ab27988, 1:250) was used. This was followed by incubation with corresponding secondary antibodies (Alexa Fluor 546- or 488-conjugated anti-rabbit or anti-mouse IgG(H+L) (A11005, Thermo Fisher Scientific, Waltham, MA, USA; 1:2000). Nuclear counter-staining was performed with DAPI (Sigma), and slides were mounted in Mowiol 4–88 (Roth). The slides were analysed by fluorescence microscopy (Zeiss Observer, Z1 microscope, Carl Zeiss, Zeiss, Germany).

### 4.4. Metabolic Activity and Viability Staining

The tetrazolium hydroxide salt (XTT) assay was used to assess the metabolic activity of A549 cells printed in the alginate/gelatin/Matrigel bio-ink at the relevant time points, according to the manufacturer’s instructions (AppliChem, Darmstadt, Germany). After the addition of the XTT reagent (1 mg/mL), the mixture was incubated for 4 h at 37 °C and 5% CO_2_. The absorbance of the resultant solution was measured spectrophotometrically at A450 nm, with a reference of A620 nm (TriStar Multimode Reader LB942, Berthold Technologies, Bad Wildbad, Germany). The lysis control consisted of cell-laden constructs cultured in culture media containing 10% Triton-X-100 (ROTH, Germany). The results were compared to lysis controls to ensure that they were comparable.

Cell-laden constructs printed in 3D were stained with 2 M calcein-AM and 4 mM ethidium homodimer-1 diluted in 1x HBSS (Thermo Fisher Scientific, Dreieich, Germany). After 30 min (37 °C, 5% CO_2_), fluorescence microscopy (Zeiss Observer) was used to analyse the samples, as described previously [34].

### 4.5. Western Blotting

The protein purification from control and irradiated 3D printed A549 samples was performed using a NucleoSpin TriPrep Mini kit, according to the manufacturer’s instructions (Macherey-Nagel, Düren, Germany). Western blotting was performed, as previously described (Al-Zeer et al., 2019) [35]. In brief, equal amounts of protein lysates were subjected to 12% sodium dodecyl sulfate-polyacrylamide gel electrophoresis (PAGE) and immunoblotted for anti-γH2AX (phosphor S139) (Abcam, ab11174, 1:200) and anti-actin (Sigma, Cat No. A2228, 1:2000). Goat anti-rabbit and goat anti-mouse IgG peroxidase conjugated secondary antibodies were purchased from Thermo Scientific. Images were visualized using ECL substrate (Pierce, Cat No. 32,109, ThermoFisher) and the ChemiDoc™ MP Imaging System (Bio-Rad Laboratories, Hercules, CA, USA), followed by densitometric analysis using Image Lab Version 4.1 (Bio-Rad).

### 4.6. Statistical Analysis

The Student’s *t*-test was used to assess the statistical significance of the experiments (GraphPad Prism 6, GraphPad Software, Inc., La Jolla, CA, USA). Data are represented as mean ±  SEM; *p*-values are considered significant at * *p* ≤ 0.05; ** *p* ≤ 0.01; *** *p* ≤ 0.001.

## 5. Conclusions

The results of this study suggest that, if more analysis methods can be modified for the use in 3D bioprints, bioprinted tumour samples, in combination with experimental radiotherapy, could become a valuable tool in the field of radiobiology research.

## Figures and Tables

**Figure 1 ijms-23-09951-f001:**
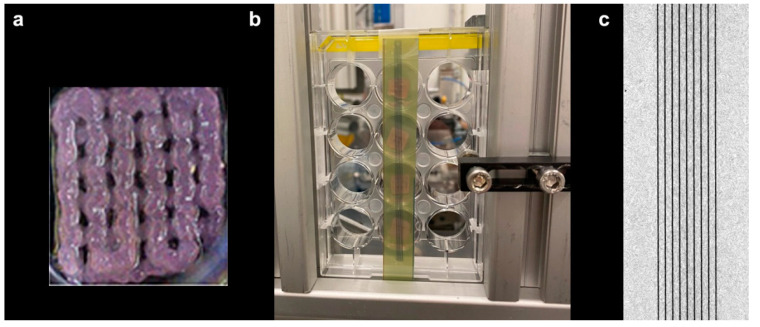
The 3D bioprinted sample (**a**) mounted for irradiation vertically in a 24-well plate (**b**). The dark stripes, caused by the peak doses of MRT, are visible on the self-developing Gafchromic™ film placed on the back of the 24-well plate, behind the samples, to record the irradiation pattern. Microphotograph of the Gafchromic™ film after irradiation (**c**).

**Figure 2 ijms-23-09951-f002:**
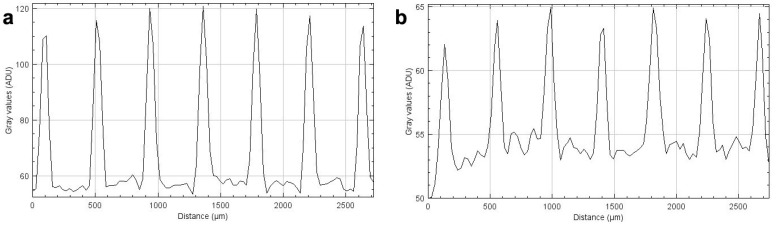
Horizontal intensity profiles form MBI peak doses of 40 Gy (**a**) and 400 Gy (**b**), obtained from HD-V2 Gafchromic™ film scans.

**Figure 3 ijms-23-09951-f003:**
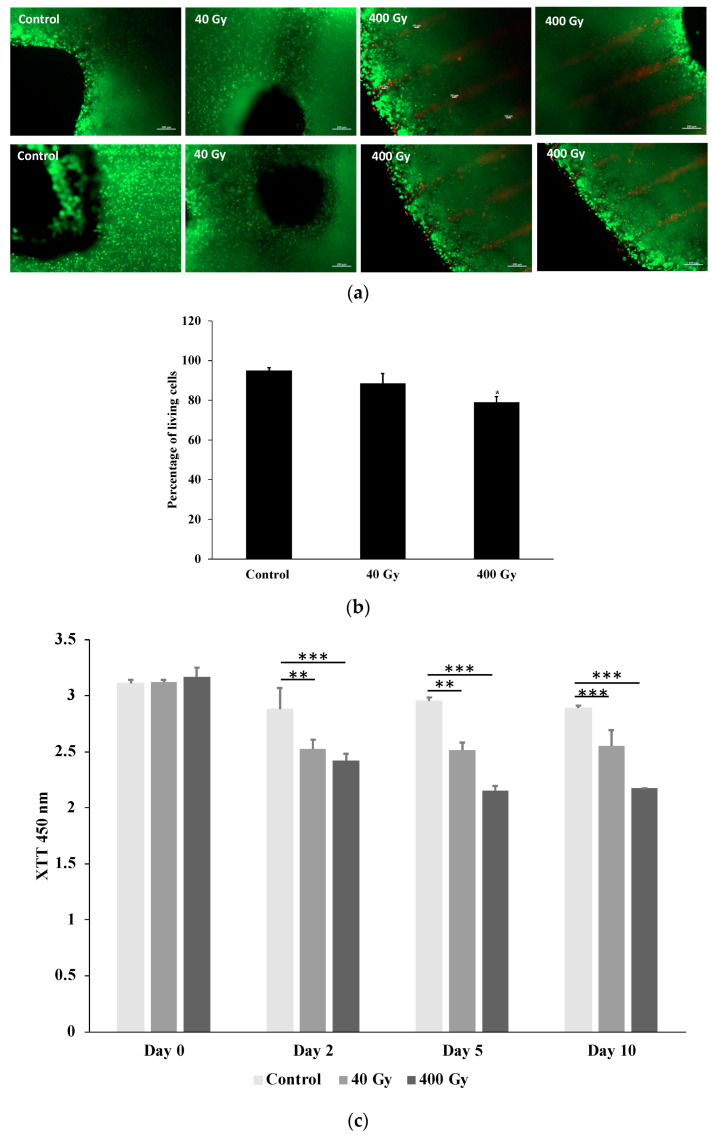
Microscopy image of live/dead stain (**a**), quantification of living and dead cells (**b**), and metabolic activity of the printed A549 cells (**c**). (**a**) Qualitative viability staining of living and dead A549 printed cells in the constructs at 48 h after irradiation using calcein-AM (live shown in green) and ethidium homodimer-1 (dead shown in red). Scale bar: 200 µm. The staining profile in the microscopy images mirrors the beam profile recorded on the film. (**b**) The estimated percentages of living and dead cells quantified using ImageJ. (**c**) The XTT test was used to assess the metabolic activity of A549 printed cells after exposure to different doses of radiotherapy at the relevant time points. Values were calculated as X-fold induction of lysis control. Data are presented as mean value ± SD; *n* = 3; * *p* < 0.05; ** *p* < 0.01; *** *p* < 0.001.

**Figure 4 ijms-23-09951-f004:**
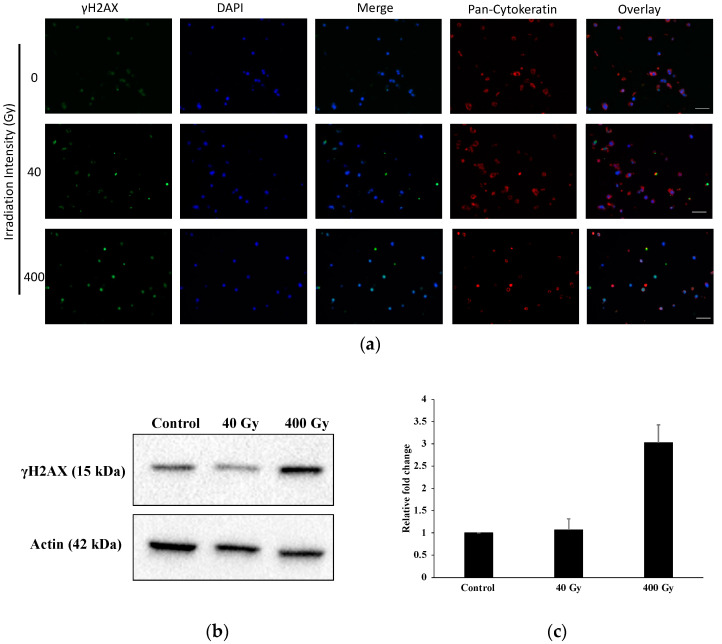
Bioprinted A549 cells immuno-stained against γH2AX following irradiation with an array of microbeams (**a**) and results of the immunoblot (**b**,**c**). (**a**) Shown are 10 µm sections of hydrogel-based 3D matrices generated from A549 lung cancer cells, fixed at 36 h after irradiation with peak doses of either 40 Gy or 400 Gy, as indicated. The γH2AX staining (green channel) indicates DNA double-strand breaks. The samples were also stained with an antibodies against pan-cytokeratin to confirm their character as epithelial cells (red channel). DAPI was used for nuclear counterstaining (blue channel). Scale bar: 50 µm. (**b**) Control and irradiated (40 and 400 Gy) samples were harvested 36 h after irradiation and subjected to immunoblot analysis for γH2AX and β-actin. The results are representative of two independent experiments. (**c**) The band densities from (**b**) were quantified and normalized to corresponding band densities of the β-actin loading control. Alterations in expression levels in the samples irradiated with MBI peak doses of 40 Gy and 400 Gy, compared to non-irradiated controls, are represented as mean fold change ± SD, *n* = 2.

## Data Availability

The data that support the findings of this study are available on request from the corresponding authors.

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
