# Peer review of "Evaluating the Suitability of 3D Bioprinted Samples for Experimental Radiotherapy: A Pilot Study"

_ijms, 2022, doi:10.3390/ijms23179951_

Round 1

Reviewer 1 Report

The article propose some premises, however the results should be considered as a very preliminary that need much more experiments to provide a constructive conclusion. Especially, that the spectrum of the paper focused on a new irradiation concepts evaluated in vitro but any ethical approval number was provided. The title of the study suggests achieving breakthrough results for applicability of 3D bioprinted samples for experimental radiotherapy, then the paper was limited into one NSCLC cell line where only 3 conditions (including control) were tested. The results provided in that way may overestimate the conclusion. Despite on focusing on NSCLC authors do not provide any references about application of the radiotherapy regarding the TNM and NSCLC staging that is crucial for this subtype of cancer. In molecular analysis I would suggest to test how irradiation methods impacted on EMT, that would be extremely important especially for early stages on NSCLC. In the end there are not indicated clear ways how the experimental 3D printing could be implemented into clinical routine. Moreover, the resolution of all the Figures is in a very bad quality.  

Reviewer 2 Report

The manuscript "Evaluating the suitability of 3D bioprinted samples for experimental radiotherapy" addresses an important issue: the need for the development of standardized samples to evaluate new therapeutic approaches and compare their efficiency with regard to tumour control. The authors conducted a pilot project to evaluate the suitability of standardized samples generated from 3D printed human lung cancer cells in radiotherapy studies.

The objectives were clearly stated and explained in the manuscript, however the experimental strategy raises some major concerns and so the experimental information from which the conclusions were drawn. The manuscript is overall well written and has good organization. Quantitative analysis of the experimental data is missing throughout the manuscript and the interpretations of the results and the discussion are thus suffering from these limitations.

The paper is interesting but there is a need for more experimental detail in order to critically review the data. Specifically, they should provide information for the following questions and comments:

Major points:

1.     It is suggested to carry out a more current review of the literature incorporating bibliography in the Introduction as well as in the Results sections.

2.     More detailed information is needed about how the tumour microenvironment was mimicked and comments on how it differs from other studies.

3.     There should be more information on why the 3D grid-like structure was chosen to model lung cancer and if the authors think the shape may influence the results by any means.

4.     Microwell plates used should be specified. Concerns over the size of the samples used and the size of the wells raise when reading the results section and there is no clarifying point through the paper on this subject.

5.     Figure 1b lacks of sufficient quality. High quality photos are requested and the caption does not adequately explain the Figure.

6.     In line 130 to 131 the doses chosen are compared to a previous experiment which is not cited and raises questions about how it does compare with the current experiments. This issue should be addressed in more detail.

7.     In section 2.3 cell viability needs more information on how it was assessed and a more quantitative method to determine it is urged.

8.     Caption in Figure 3 is scarce and a more detailed description is needed.

9.     Figure 4 should be redone joining panels A and B and better describing the caption. 3 replicates per experimental group is the standard, so it should be done this way or comment and justify why it is not done as the standard carried in similar studies in the field.

The molecular weight marker has to be shown in the immunoblots. The images of the gel have to be labeled otherwise evaluation is impossible, for example what is the fourth sample in the first and second gel and why it is not discussed in the paper?

Quantification of the signal of the bands is also recommended as done in other similar studies.

Minor points:

1.     In line 249 “in vivo” should be italicized.

2.     Figure 2 A and B labels are distorted and should be added at the top not at the bottom of each panel. The same applies to Figure 1.

3.     Figure 3 seems like the lines of the statistical analysis have different length and are not centered. In this Figure there are no labels to panels A and B, add them please. The scale bar is also barely visible in panel A.

Round 2

Reviewer 1 Report

Authors have responded to all my and other reviewer remarks and improved the quality of the manuscript. I would suggest some minor changes before publishing:

1. Please note that some figures' camption need adjusment.

2. Added lanes in the discussion paragraph (240-248) require relevand refferences.

3. The conclusin paragraph should be placed after the discussion

Author Response

I would suggest some minor changes before publishing:

  1. Please note that some figures' camption need adjusment.

Response:

DONE.

  1. Added lanes in the discussion paragraph (240-248) require relevand refferences.

Response: Reference inserted, subsequent reference numbers adapted accordingly.

  1. The conclusin paragraph should be placed after the discussion.

Response: DONE.

Many thanks for yor time and effort in reviewing our manuscript and for your final suggestions.

Reviewer 2 Report

The revised firm has been evaluated and I recommend it to be accepted in present for. 

Author Response

Response: Thank you for your time and for your kind evaluation.